# Interplay of changing irrigation technologies and water reuse: Example from the Upper Snake River Basin, Idaho, USA

Shan Zuidema[1], Danielle Grogan[1], Alexander Prusevich[1], Richard Lammers[1], Sarah Gilmore[2], Paula Williams[2]

[1]Earth System Research Center, University of New Hampshire, Durham, NH, 03824, USA
[2]Center for Resilient Communities, University of Idaho, Moscow, ID, 83844, USA

Correspondence to: Shan Zuidema (shan.zuidema@unh.edu)

**Abstract.** Careful allotment of water resources for irrigation is critical to ensuring the resiliency of agriculture in semi-arid regions, and modernizing irrigation technology to minimize inefficient losses is an important tool for farmers and agricultural economies. While modernizing irrigation technology can achieve reductions in non-beneficial use of water such as bare soil evaporation, non-consumptive losses or water returned back to the landscape are also reduced, often eliminating flowpaths that other users rely on. In basins using a combination of surface and groundwater, replenishing aquifer storage by the managed aquifer recharge (MAR) of seasonally available water can mitigate the aquifer drawdown that results from reduced recharge when irrigation efficiency is improved. We examine the effects of MAR on the system-scale efficiency of modernizing irrigation technology and the resulting changes to the reuse of non-consumptive losses using a macro-scale hydrologic model applied to the semi-arid Upper Snake River Basin (USRB) of western Wyoming and southern Idaho, USA. Irrigation technologies were represented explicitly in the model, and available data informed baseline parameterizations of irrigation technology. A suite of parameterizations were simulated that updated existing technologies to be more efficient, both with and without sufficient MAR to cause stabilization of the aquifer at present-day head. As expected, simulated changes to irrigation technology resulted in greater downstream export of pristine water and a higher rate of aquifer drawdown when MAR was not simulated. Under current water use and cropping patterns, we were not able to simulate aquifer stabilization and maintain discharge downstream at any level of irrigation efficiency. We found support for the hypothesis that as efficiency improves, less MAR is required to maintain a stable aquifer than returns flows are reduced due to increased efficiency. To evaluate the hypothesis, we defined the management benefit as a metric that compared the difference between the change in irrigation's net recharge from the change in MAR required as irrigation technology became more efficient. The metric generally indicated that less MAR was needed than net recharge was lost, but only for the most efficient case did the management benefit exceed the MAR needed at baseline to stabilize the aquifer. Increasing efficiency of irrigation technology reduced reuse, the gross irrigation derived from prior non-consumptive losses, but simulating MAR increased reuse for a given parameterization, leading to higher effective irrigation efficiency. We find that local groundwater storage that users depend on is generally more sensitive to management decisions than downstream flows, and drawdown of the aquifer without MAR always exceeded any decrease in discharge induced by MAR. Improving resource sufficiency in semi-arid systems like the USRB will require an array of solutions that will necessarily weigh benefits to local and downstream users.

# 1 Introduction

Access to irrigation water is critical to determining the future resiliency of many agricultural systems (Foley et al., 2011), and challenges of providing irrigation water require close scrutiny of its efficient use (Grafton et al., 2018). The goal of resilient agricultural systems should reflect a global need to reduce water scarcity (Rosa, 2017), with adaptations that are often context specific (Vanham et al., 2018). Successful management of water resources to protect against water scarcity requires consideration of the specific interactions of multiple actors (Keller and Keller, 1995).

One suite of solutions where water is scarce is to modernize irrigation technology to ensure that the greatest proportion of supplied water is used for beneficial crop growth (Gleick et al., 2011; Jägermeyr et al., 2015, 2016). Improving classical irrigation efficiency (CIE), the ratio of beneficial consumptive use to gross irrigation abstractions, is critical to meet agricultural production needs (Jägermeyr et al., 2016), and additionally has myriad co-benefits such as reduced energy use or improved water quality (Gleick et al., 2011; Vanham et al., 2018). However, more efficient irrigation systems tend to counterintuitively increase total water consumed, or at least do not decrease use to the degree expected. As efficiency increases, usually at a cost to the irrigator, the water available from reduced losses can be applied for higher value and more water intensive crops (rebound) or for expanding crop areas (slippage) (Contor and Taylor, 2013; Grafton et al., 2018; Pfeiffer and Lin, 2014; Tran et al., 2019), especially when users are encouraged to extract a full water allotment by legal doctrines such as the Prior Appropriations system used in the US West. Increasing CIE also tends to reduce non-consumptive losses that downstream users rely on (Foster and van Steenbergen, 2011; Frederiksen and Allen, 2011; Grafton et al., 2018; Grogan et al., 2017; Simons et al., 2015). Non-consumed losses, the fraction of water applied by irrigation that flows back to the landscape, follow different pathways. The term *irrigation returns* can refer to flow or flow structures conveying non-consumed water off irrigated fields and back to a canal system (e.g. Lin Y. and Garcia L. A., 2012), percolation back to a source aquifer (e.g. Dewandel et al., 2008), or more generally to all water not consumed by irrigation water application or delivery (Grogan et al., 2017; Keller and Keller, 1995; Simons et al., 2015); we adopt the latter meaning when referring to *irrigation* or *incidental returns*.

Investigators of water resources argue that the reuse of incidental returns increases the basin or global efficiency of supplied water, making technological investments that increase CIE less effective when considered at basin-scales rather than at farm-scales (Keller et al., 1996). The effect has been observed empirically in well-studied basins (Simons et al., 2015). Increasing CIE is almost certainly a critical component to maintain the resiliency of agricultural systems when only surface or groundwater supplies irrigation and will necessarily reduce the incidental return back to the system. In settings that conjunctively use both surface and groundwater resources, managed aquifer recharge (MAR) can increases the adaptability and resiliency of irrigated agriculture (Dillon et al., 2020). MAR adds water to aquifer storage when available, eliminates the need for infrastructure associated with surface reservoirs, minimizes surface evaporation, and can be less expensive than surface storage (Arshad et al., 2014; Dillon, 2005; Dillon et al., 2019; Maliva, 2014; Scanlon et al., 2016). MAR as part of a conjunctively managed water resource system has been demonstrated to maintain water supplies for irrigated agriculture during drought (Foster and van Steenbergen, 2011; Guyennon et al., 2017; Niswonger et al., 2017; Scanlon et al., 2016; Tran et al., 2020). However, water used for MAR tends to reduce flow leaving a

catchment (Yaraghi et al., 2019), which may have important downstream consequences. In other cases, MAR may affect annual flows slightly (e.g. Niswonger et al., 2017), but can shift timing of baseflow entering rivers from the aquifer to summer months, providing important temperature refugia for aquatic species (Van Kirk et al., 2020).

Despite the potential benefits from coupling MAR with conjunctively managed water sources, there remain challenges in uptake of the practice (Dillon et al., 2020) to address globally declining aquifer storage (Bierkens and Wada, 2019). Outside specific regulatory intervention, the practice of MAR can marginally reduce the cost to pump groundwater such that MAR would be expected to result in rebound and slippage effects (Tran et al., 2019) where more land is planted, or more water intensive crops are grown to utilize the available water. Benefits from the conjunctive management of water resources and MAR are projected to be greater in arid environments (Scanlon et al., 2016).

The two interventions presented above, increasing the efficiency of irrigation through technological modernization and MAR appear to synergistically alleviate the drawbacks of each practice. In the absence of slippage, increasing CIE can reduce incidental recharge (Simons et al., 2015), but retains greater flow within the river, whereas MAR increases recharge but reduces annual river flow (Yaraghi et al., 2019). Balancing the two interventions could potentially achieve greater resiliency of irrigated agriculture than either alone. To date there have been limited analyses to include both strategies in the same framework. Tran et al. (2019) account for specific efficient irrigation practices within the context of multiple potential drivers in an hydro-economic analysis. Other examples of mechanistic models applied to the problem of MAR and conjunctive resource management have assumed static efficiencies of irrigation technology and crops (Niswonger et al., 2017; Scherberg et al., 2014). Here we consider the coupled influences of irrigation technology modernization and MAR on water resources to assess the limits to which either intervention could achieve aquifer stabilization, while maintaining downstream flows above critical thresholds.

We quantify the impact of changing irrigation efficiency on basin water stocks, aquifer recharge, downstream discharge, and within-basin water reuse using the test case of the Upper Snake River Basin (USRB) of Idaho, USA, an intensive agricultural setting in the semi-arid American west (Figure 1) that relies on both surface and groundwater. Arid and semi-arid agriculture can be very important economically; 31-36% of the nation's net farm income is produced in arid or semi-arid regions (Trabucco and Zomer, 2019; USDA NASS, 2014). Historic flood irrigation of river water elevated aquifer head above pre-irrigation levels, and in the latter half of 20th century aquifer storage has declined (Kjelstrom, 1995) due to increasing groundwater pumping and decreasing recharge of surface water as irrigation technology has modernized. Therefore, aquifer stabilization is critical for establishing resilience of the agricultural system (IWRB, 2016). Implementation of recharge (as MAR) and other management actions to ensure a resilient agricultural system in the USRB may provide important insights relevant throughout arid and semi-arid regions where surface evaporative fluxes are similarly high (Carr G. et al., 2010; Ghassemi et al., 1995; Tal, 2016).

Careful application of improved irrigation efficiency and MAR has been part of on-going management strategies in the USRB. Water is governed in Idaho under the Doctrine of Prior Appropriation, which allocates water to users according to the date when they first put water to continuous beneficial use. As aquifer drawdown has continued, aquifer recharge, water transfers, and other water conservation efforts have been classified as beneficial uses (Fereday et al., 2018). Water users have self-organized in an effort to stabilize the aquifer and optimize use of the water within the USRB, and have moved to a more conjunctive management

of surface and groundwater resources (Gilmore, 2019). A moratorium on ground water permits (Higginson, 1992) and conservation efforts have resulted in a reduction in consumptive ground water use, and motivated the adoption of targets for 0.3 km$^3$ y$^{-1}$ of MAR as an intervention (IWRB, 2016); however, aquifer storage may still be declining. Simultaneously, maintaining sufficient downstream flow from the basin is strictly required for senior water rights holders and hydropower generation (IWRB, 1985). The

USRB is an ideal setting to assess the trade-offs between within-basin aquifer storage and downstream supply through conjunctive management.

In this study, we frame a series of model parameterizations together to test hypotheses guided by the key constraints of water resource management in the USRB. We utilized a distributed model of hydrologic function and human water use to estimate the recharge required to a) stabilize the aquifer under present-day irrigation efficiencies, and b) offset reduced irrigation returns

from continued modernization of irrigation technology. We performed simulations introducing progressively more efficient irrigation technology to a baseline representation of the USRB, which required reduced withdrawals from the Snake River, but which hastened aquifer drawdown by decreasing recharge of incidental returns. These simulations were paired with counterparts introducing sufficient managed aquifer recharge to ensure negligible change in aquifer storage (stabilization) over the same period. We hypothesized that only a fraction of the reduced incidental returns from modernizing technology would be needed to maintain

aquifer volume if introduced as MAR. An alternative hypothesis is that asynchronicity in recharge water availability and irrigation demand, coupled with fairly fast flow through the aquifer system, would require greater recharge rates than if water was introduced as inefficient irrigation and reused contemporaneously. For each simulation we calculated the total amount of previous incidental returns reused as gross irrigation, using the model's core capability of tracking water sources through all pools of the hydrologic cycle. We hypothesized that simulations with additional MAR would exhibit lower reuse than simulations without MAR because

a greater proportion of recent snowmelt would recharge the regional aquifer. Alternatively, additional MAR reduces surface water availability and may promote groundwater abstractions that would favour greater reuse as most irrigation returns percolate to recharge the aquifer.

## 2 Methods and Data

The following sections describe the setting (Sect. 2.1), describe the formulation of hydrologic fractions used here (Sect. 2.2), the

experiments conducted (Sect 2.3), the Water Balance Model (WBM) – a distributed hydrologic model representing anthropogenic water uses (Sect. 2.4), model input data (Sect. 2.5), and validation criteria (Sect. 2.6).

### 2.1 Upper Snake River Basin

The Upper Snake River Basin (USRB) is a semi-arid steppe ecosystem with a snow-melt dominated Mediterranean climate in western Wyoming, and southern Idaho, USA (Figure 1). The 92,700 km$^2$ basin is bounded to the east by the Teton Mountains, and

to the north by the Sawtooth and Bitterroot Mountain ranges. Precipitation over the Snake River Plain is generally less than 250 mm/year but averages about 400 mm/year (or 46.3 km$^3$/year) over the whole basin with most water entering the river network as

montane snowmelt. The basin is underlain by Quaternary basalts of the Snake River Group (Whitehead, 1992), which form the highly transmissive Eastern Snake Plain Aquifer (ESPA). Irrigation in the USRB began in the late 1800s and gravity drained flood irrigation was the primary mode of irrigation until the mid-1900s (Lovin, 2002; Wulfhorst and Glenn, 2002). Incidental recharge from the non-consumptive losses of irrigation water increased storage in the ESPA, and increased discharge from a dense collection of springs in the Snake River canyon between Milner and King Hill, Idaho (Kjelstrom, 1995). Through the latter half of the 20th century, aquifer head declined due to increasing reliance on groundwater for irrigation and reduced incidental recharge (Moreland, 1976) as flood-irrigated land transitioned to sprinklers. Aquifer stabilization at today's head is a primary concern in the basin, even though head is above pre-irrigation levels. State agencies are practicing managed aquifer recharge (MAR), the deliberate infiltration of seasonally available water for use throughout the year, as one technique in the conjunctive management of water resources (IWRB, 2009, 2016).

Groundwater age dating and geochemical analysis established that the downgradient portions of the aquifer consist of between 60 and 80% of water used for irrigation and derived from the Snake River (Lindholm, 1996; Plummer et al., 2000). A highly managed network of reservoirs and canals convey about 12 km$^3$ y$^{-1}$ of water to croplands (Maupin et al., 2014), equivalent to about 25% of annual precipitation to the basin. At least 5.5 km$^3$ of water is stored in the three largest reservoirs alone. An additional 2.5 km$^3$ y$^{-1}$ is abstracted from the ESPA by irrigators (Dieter et al., 2018; Maupin et al., 2014), and approximately 5 km$^3$ y$^{-1}$ of water returns from the ESPA to the Snake River through a series of springs (Covington and Weaver, 1991; Kjelstrom, 1995). Inflows to the ESPA include several losing rivers at the northern extent of the USRB and the Snake River, which loses water directly to the ESPA near American Falls reservoir (Lindholm, 1996; McVay, 2015). Spring flows out of the ESPA are critical for maintaining an aquaculture industry along the Snake River canyon, and constitute a majority of Snake River discharge out of the USRB supporting critical aquatic habitats, hydroelectric generation potential, and irrigation of downstream agriculture. Water available from the Upper Snake River and the ESPA provide irrigate numerous agricultural products with dairy forage, beet sugar, and potato being the most economically-important (USDA NASS, 2014).

## 2.2 Hydrologic fractions and irrigation resource use

Defining efficiency of agricultural water use is complicated because water lost non-productively by one water user may be used productively elsewhere downstream in the basin, making terms describing efficiency or resource sufficiency specific to the spatial scale considered. We describe irrigation efficiency using hydrologic fractions that describe the fate of water abstracted from either surface or groundwater sources for the purpose of irrigation (Frederiksen and Allen, 2011; Haie and Keller, 2008; Lankford, 2012; Perry, 2011). Water abstracted as gross irrigation ($G$) can have three fates when added to irrigated pixels at the plot-scale: i) beneficial use ($B$) is the irrigation water used for beneficial crop growth; ii) non-beneficial consumption ($N$) is water evaporated non-beneficially from soil or canals; or iii) non-consumptive loss ($L$, herein *incidental returns* or *incidental recharge*) is runoff or percolation as a liquid that remains in the basin (Figure 2). Of these plot-scale fates of gross irrigation water, $B$ and $N$ are both assumed terminal because liquid water leaves the domain as vapor or in crops. Incidental returns ($L$) on the other hand remain in the system and fates at the basin scale include export ($X$) via streamflow at the basin outlet, evaporation ($E$) from the surface water

network, human use ($U$), reuse as gross irrigation ($R$), and net storage ($S$) primarily in the aquifer; however net storage in surface reservoirs, and soil is also calculated.

WBM tracks key component volumes, including incidental returns ($L$), to all terrestrial compartments of the hydrologic system permitting direct computation of gross irrigation water reuse ($R$). In our analysis we assume that all incidental returns are recoverable, and therefore do not make a distinction between recoverable and non-recoverable returns (as in Lankford, 2012), and directly assess the volumes of water *recovered* in gross abstractions. Gross irrigation reuse ($R$) is the weighted sum of abstractions consisting of incidental return in each source and is calculated daily Eq. (1).

$$R_{i,j} = I_{Aqf} \cdot f^{irr}_{Aqf_{i,j}} + I_{Rsvr} \cdot f^{irr}_{Rsvr_{k,l}} \tag{1}$$

where $i$ and $j$ are row and column indices for the point of irrigation water application; $I_{Aqf}$ and $I_{Rsvr}$ are the abstracted irrigation water from aquifer, and surface reservoirs, respectively; $f^{irr}_{Aqf}$ and $f^{irr}_{Rsvr}$ are the fraction of irrigation return flow water in aquifer and reservoir water, respectively; and $k$ and $l$ are row and column indices for the pixel of the surface supply reservoir. The metric was summed spatially and temporally and compared to total gross irrigation ($G$) to calculate a ratio of irrigation water reused within the USRB. Irrigation efficiency is calculated as classical irrigation efficiency (CIE) given by Eq. (2), and effective irrigation efficiency (EIE) following the quantity (Type N) model of Haie and Keller (Haie and Keller, 2008) given by Eq. (3).

$$CIE = \frac{B}{G} \tag{2}$$

$$EIE = \frac{B}{G-R} \tag{3}$$

Note, $R$ does not quantify how many times a given parcel, or on average all irrigation water, is reused, as in the distinct definition of $\boldsymbol{R}$ as the index of unsustainable groundwater reuse in Grogan *et al.* (2017). Rather, it identifies what portion of total irrigation water has been through cycles of use (Figure 2).

## 2.3 Experiment structure

There is strong connection between the Upper Snake River and ESPA in the USRB, both through reach gains and sinks from the Snake River to the ESPA and from springs back to the Snake River. These connections are not unlike alluvial aquifers where conjunctive management of water resources is most common (Foster and van Steenbergen, 2011). We therefore use predictive inference (Ferraro et al., 2019) to assess the potential for trade-offs between downstream flow and aquifer drawdown as irrigation efficiency and MAR change independently. We should note that the experiments we perform potentially violate water law and precedent in the basin (Gilmore, 2019), so natural experiments (Penny et al., 2020) to interrogate similar processes are impractical. To test our hypothesis that only a fraction of reduced incidental recharge is needed as managed aquifer recharge (MAR) to increase water availability basin-wide, we simulate a suite of alternative model parameterizations to capture increasing irrigation efficiency (as CIE) paired with and without MAR. In WBM, we introduce a fraction of daily flow from the Snake River immediately above the American Falls Reservoir directly to the ESPA to represent recharge as an intervention. Because our simulations also

reflect changes in aquifer recharge related to changing flow in the river source, we use the term enhanced aquifer recharge (EAR) to refer to all induced changes in aquifer recharge in our model simulations. Specific changes to simulated irrigation technologies for each parameterization are described below. We then assess our hypothesis by calculating a management benefit (MB) metric that compares the change in net incidental recharge to the change in EAR required for aquifer stabilization by difference for each parameterization. The MB is the magnitude by which the increase in required EAR is less than the loss in net incidental recharge and calculated by Eq. (4).

$$MB = (I_{rch}^* - I_{rch}) - (EAR - EAR^*) - \frac{dV_{ESPA}}{dt} \tag{4}$$

where $I_{rch}$ is net incidental recharge (incidental recharge minus groundwater abstraction), $EAR$ is the enhanced aquifer recharge flux, and $^*$ represents the flux at the present-day baseline. For each parameterization, we compare the change in aquifer storage with the relative change in discharge from baseline to evaluate the combination of aquifer and streamflow capture needed to support irrigation abstraction at a given level of efficiency. In this manuscript, our definition of streamflow capture is general, any decreasing discharge out of the basin due to altered management practice, and does not specifically mean the change in streamflow and recharge resulting from increased groundwater pumping (Konikow and Leake, 2014).

For each simulated suite of irrigation technology parameters, we run paired simulations with and without EAR. For EAR simulations we target aquifer stabilization defined as a long-term (e.g. decadal) average change in groundwater storage of the entire ESPA close to zero during the contemporary time-period from 2008 through 2017 Eq. (5).

$$\frac{dV_{ESPA}}{dt} \sim 0: -0.1 < \frac{dV_{ESPA}}{dt} < 0.1 \text{ [km}^3 \text{ y}^{-1}] \tag{5}$$

Once values for ESPA exchange were calculated for the baseline representation, simulations were conducted with these values for each of the nine more efficient irrigation technology parameterizations (Table 1). Then, additional EAR was estimated through manual calibration to achieve a stabilization of ESPA volume for the baseline and each of the efficiency parameterizations. For all model simulations, aquifer stabilization, basin discharge, and hydrologic fractions including reuse were calculated from hydrologic model output. In calculating MB, the change in aquifer volume is subtracted to account for small deviations from aquifer stability that remain after calibration.

**2.4 Water Balance Model**

We used the University of New Hampshire Water Balance Model (WBM) to characterize water balance and assess water resource fates (Vörösmarty et al., 1989; Wisser et al., 2010). WBM is a distributed hydrologic model utilizing conceptual soil, surface runoff, and shallow groundwater pools, a one-dimensional river network utilizing hydrologic routing schemes, and representations of human controls on the hydrologic cycle such as dams, impervious surfaces, irrigation, livestock, industrial, and domestic water use. WBM tracks specific components of water fluxes, notably irrigation returns, through each represented pool assuming each pool is well-mixed at each daily time-step (Grogan et al., 2017).

Several modifications were implemented in WBM for the present work; a more complete description of the fundamental WBM model structure is available elsewhere (Grogan, 2016; Grogan et al., 2017; Wisser et al., 2010). In previous applications of

water tracking in WBM (Grogan et al., 2017), component stocks were adequately cycled as representative of the various components following model spin-up. To address concerns that water components retain a memory of assumptions at initialization owing to new groundwater representation described below, all stored water at model initialization was tracked as *relict* water, a measure of water remaining in the system prior to the dynamic model simulation epoch. We introduced an upper volumetric bound to the surface runoff pool to rectify a low bias in runoff during extreme precipitation and snowmelt events. The fraction of surplus soil water (soil water-content above field capacity) that flows to the shallow groundwater pool ($\gamma$,unitless), and it's complement (1-$\gamma$), which is directed to the surface runoff pool, are generally about 0.5 and robust in a range from 0.4 to 0.6 (Grogan et al., 2017; Samal et al., 2017; Stewart et al., 2013; Zuidema et al., 2018). Due to the highly permeable geology found along the Eastern Snake Plain, $\gamma$ was increased to represent high initial infiltration rates common throughout the Eastern Snake Plain (IDWR, 2013). For our simulations, $\gamma$ was spatially variable (ranging from 0.38 to 0.96, mean = 0.73) based on elevation as a proxy for the extents of the Eastern Snake Plain (Figure S1). Other parameters defining the major hydrologic controls were established by work across multiple scales (Grogan et al., 2017; Samal et al., 2017; Stewart et al., 2013; Wisser et al., 2010; Zuidema et al., 2018) and were not calibrated for this application in the USRB.

Several features were added to WBM to implement the experiment. To represent the intense management of USRB water resources, reservoir outflow from the three largest reservoirs were specified; therefore, WBM predicted reservoir volume as a consequence of managed release. Irrigation technology was revised in WBM to a process-based representation that redistributes inefficient irrigation water via surface runoff flows, groundwater percolation, and evaporation during both delivery and application stages. The system explicitly represented non-beneficial consumption as evaporation of sprinkler mists and evaporation from canal and soil surfaces, using technology specific parameters reflecting county-wide averages from USGS water use statistics (Dieter et al., 2018; Maupin et al., 2014). A representative fraction of 4% of sprinkler applied water is evaporated as mists (Bavi et al., 2009; McLean et al., 2000; Uddin et al., 2010). Further, during the irrigation season, water is assumed to be evaporating at potential rates throughout the canal network. We assume crop ET is required (i.e. beneficial) for both transpiration and salt flushing, but water applied during an irrigation event in excess of daily crop demand wets soil above field capacity. . Incidental losses during application followed Jägermeyr et al. (2015) and we used their estimates of the distribution uniformity parameter that prescribed excess water needed to satisfy net irrigation demand based on the type of technology, either drip, sprinkler, or flood. Excess water evaporates (non-beneficially) at the potential rate, and unevaporated water is returned non-consumptively at the end of the timestep via either percolation, or runoff if vertical hydraulic conductivity is too low. The algorithm describing irrigation water fates is detailed in the Supplemental material.

We defined surface water sources for each administrative basin in Idaho (IDWR, 2015) to come from one or more reservoirs based on the canal network's distribution (Figure S2). All daily surface abstractions for irrigation are made from the pool of source reservoirs providing water to each administrative basin proportional to their available storage. Groundwater abstractions for irrigation of croplands were calculated as the difference in demand not satisfied by surface water sources. A more detailed aquifer representation was needed here than in previous WBM studies. We simulated the ESPA over the same extents as the ESPAM2.1 model (IDWR, 2013) using a lumped formulation that received distributed recharge from natural and incidental sources and reach

gains from specific losing rivers (Figure 1), provided a pool of groundwater available for irrigation, and discharged to a series of 213 springs along the Snake River canyon (Covington and Weaver, 1991). Discharge from springs was head dependent and sub-daily head and outflow were calculated numerically using a third order scheme (Bogacki and Shampine, 1989). We represented the aquifer as upgradient (northeast) and downgradient (southwest) lumped compartments (Figure 1) to reflect two characteristic types of water identified by Plummer *et al.* (2000), old groundwater in the upgradient portion, and young water derived from incidental recharge of Snake River water in the downgradient or southwest portion. Storage parameters were estimated for each section: upgradient specific yield is 0.06 and thickness of the aquifer is 250 m; downgradient specific yield is 0.05 and thickness is 220 m (Garabedian, 1992; IDWR, 2013; Whitehead, 1992). We represented the hydraulic connection between the ESPA and the American Falls Reservoir (Garabedian, 1992; IDWR, 2013) as a drain/spring pair. Additional details of the implementation of the lumped aquifer solution are presented in the Supplement.

## 2.5 Input Data

We used a topological network of the Upper Snake River Basin (USRB) that covered an area of 92,900 km$^2$ at a spatial resolution of 30-arcseconds (approximately 780-m) based on HydroSHEDS (Lehner et al., 2008) but refined to better represent drainage as mapped by the US Geological Survey's National Hydrography Data (USGS, 2019). Reservoir data was derived from the National Inventory of Dams (USACE, 2016) and updated manually to include additional dams, refine reservoir capacities, remove secondary structures on reservoirs, and refine the locations and upstream drainage areas. Reservoir outflow came from observed flow data from USGS gaging stations located immediately downstream of three primary irrigation reservoirs: gage 13011000 in Moran, WY below Jackson Reservoir, gage 13032500 in Irwin, ID below Palisades Reservoir, and gage 13077000 in Neeley, ID, below American Falls Reservoir. No data regarding direct abstractions from reservoirs were available from these sources. Additionally, we increased the total capacities represented in WBM of these three reservoirs by 10% to approximate storage of their downstream canal systems. There were 128 dams and corresponding waterbodies in the USRB domain. WBM simulations used gridMET (Abatzoglou, 2013) for contemporary precipitation and temperature and MERRA2 for open water evaporation (Gelaro et al., 2017). We utilized a temperature based evaporation equation (Hamon, 1963) for calculating potential evapotranspiration (PET) and a temperature-index based snow accumulation and melt formulation (Willmott et al., 1985). Human population density, which controls both domestic and industrial water demand, came from SEDAC Gridded Population of the World (CIESIN et al., 2016). WBM simulations used Food and Agricultural Organization (FAO) estimates of livestock density for cattle (Steinfeld et al., 2006) at 5 minute resolution following Wisser et al. (2010). These data compared favourably with USDA National Agricultural Summary Statistics (NASS) for 2005, but exhibit more realistic spatial variability than county-level averages in NASS. Over the USRB domain, NASS livestock density is approximately 2 head/km$^2$ density representing a low bias of the FAO data of less than 1%. We utilized USDA Soil SURvey GeOgraphic (SSURGO) data to parameterize available water capacity for USRB soils. We specified a rate of 115 mm/day for percolation below land occupied by canals and irrigated lands exceeding saturation following findings from the Idaho Water Resources Board (2016).

WBM adapts FAO's methodology (Allen et al., 1998) to estimate crop water requirements based on reference ET, soil moisture, crop coefficient ($k_c$) and is detailed in previous work (Grogan et al., 2017; Wisser et al., 2010). Here, we utilized the US Department of Agriculture's Crop Data Layer (CDL) estimates of crop types and land cover at 30 m resolution (Han et al., 2012) after remapping crop groups (Table S1). The proportions of irrigation delivery technologies were spatially homogenous and reflected the average lengths of technologies in the USGS National Hydrography Dataset (nhd.usgs.gov). The relative proportions of application technology varied by county following USGS surveys (Dieter et al., 2018; Maupin et al., 2014). To address our first two hypotheses, parameterizations were defined that represent nine progressively more efficient suites of irrigation technology, identified here as parameterizations Eff.A through Eff.I. The nine parameterizations are controlled by the relative fraction of flood irrigation (with corresponding increases in sprinkler area), the relative fraction of drip irrigation (with corresponding decreases in flood and sprinkler area), the fraction of canals (with corresponding increases in pipes), and the percolation factor of canal bottoms (Table 1).

## 2.6 Model validation metrics

Model assessment used a composite objective function that described model-observation misfit across four primary metrics. We compared: 1) monthly flow from the springs draining the ESPA against total gains minus diversions between the Kimberly and King Hill, Idaho USGS gaging stations provided by the IDWR (Sukow, 2011, personal comm.); 2) annual gross and surface water abstractions for irrigation over the USRB aggregated by county for the years 2010 and 2015 and compared to USGS water use statistics (Dieter et al., 2018; Maupin et al., 2014); 3) seasonal river discharge at locations upstream of actively regulated reservoirs at USGS gages 13010065 (Flagg Ranch, Wyoming), 13137500 (Trail Creek, Ketchum, Idaho), and 13039500 (Henry's Fork, Lake, Idaho); and 4) seasonal storage within the actively regulated Snake River reservoirs against data from the US Bureau of Reclamation Hydromet database. A standard suite of statistics are used to assess each of these metrics, and we report percent bias and Nash-Sutcliffe efficiency (NSE) for the period between 2008 and 2017. Manual parameter calibration established reasonable estimates for the water exchange between the Snake River and ESPA near American Falls. Exchange between the Snake River and ESPA affects reservoir volume estimates, aquifer volume (and therefore spring flows), and can affect surface irrigation estimates as abstractions are necessarily curtailed if American Falls reservoir does not have sufficient storage to meet demand. Therefore, focusing on all four metrics to establish performance was necessary.

## 3 Results

### 3.1 Model Validation

Though most processes within the model were uncalibrated, WBM accurately represented observations of the key fluxes in the USRB that we tested. The spatial distribution of abstractions is accurate for both total and surface sources of irrigation water (Figure S3). Mean annual discharge from springs draining the ESPA is unbiased (Figure S4). Interannual variability in peak runoff is generally well captured (Figure S5a), and timing of peak runoff generation from snowmelt is accurate; however the onset of

snowmelt tends towards an early bias in most years. Though discharge from the headwaters of the Upper Snake River in the Teton Mountains was well characterized (e.g. monthly discharge NSE 0.72 and bias of 13% at Flagg Ranch, WY – Figure S5a), the intense management of reservoirs in the USRB results in cascading errors in simulating the timing, rates, and magnitudes of reservoir drawdown (Figure S5). Representing the hydrology of heavily managed basins, such as the USRB where most large reservoirs are

managed as a single system (not just three reservoirs where we forced outflows to observations), with macro-scale models is challenging and development of robust representations of management of reservoir series are important directions of future research (Adam et al., 2007; Masaki et al., 2017; Rougé et al., In review).

        To address our specific hypotheses, we compared a series of model parameterizations from a common baseline. Biases in model representation of the USRB from utilizing a minimally calibrated model are common between each hypothetical

parameterization of changing irrigation efficiency. Therefore, the differences between model simulations are informative of the effect that interventions of irrigation technology have on semi-arid agricultural basins generally. Inferences specific to the USRB's response to similar management interventions are inevitable, so it is worth considering how known model misfit could influence interpretations of the fate of incidental returns, irrigation reuse, and the effectiveness of coupling EAR with increased irrigation efficiency for the USRB specifically. We note several obvious biases between the model simulation at baseline and observations.

First, WBM predicts the onset of snowmelt early in most years (Figure S5a), which leads to overfilling of the major reservoirs along the cascade of reservoirs through the Upper Snake River. Early season discharge leads to overfilling of reservoirs compared to observations, and then shunting of water downstream causing both a high-bias in early season discharge at the basin outlet (Figure S5f), and less water in storage late in the season. Excess discharge at the outlet ranges between 0.65 to 8.75 $km^3$ $y^{-1}$ with a median of 2.86 $km^3$ $y^{-1}$. Furthermore, the early shift in snowmelt makes less water available in the reservoir cascade later in the year leading

to overdraft of Palisades Reservoir late in the irrigation season in 2010, 2012, 2015, and 2016, and therefore less water available to American Falls reservoir in those years. Model simulations that more accurately captured the timing of snowmelt onset with the known reservoir management would retain more snowmelt in the reservoir cascade making more snowmelt available to maintain reservoir levels near observations, and for irrigation. Therefore, less groundwater would likely be used for irrigation resulting in less aquifer drawdown and a lower rate of gross irrigation reuse of incidental returns.

We also note that seasonal dynamics of the water table and therefore discharge through springs were highly damped relative to observations, which results from our lumped aquifer parameterization. Prior analysis shows that annual cycles in spring discharge results from fluxes that occur within 20 km of the springs (Boggs et al., 2010). Though mean spring discharge is unbiased, incidental recharge to the aquifer and pumping from the aquifer are spread over the two compartments of the aquifer and exceed the space scales that would create seasonal dynamics. Suppressed seasonality of spring discharge could reduce seasonality of downstream

flows; however, there are no major abstractions of surface water downstream of spring in our representation of the USRB. Moreover, seasonal head fluctuations could reduce pumping by either drying wells, or increasing pumping costs; however, these dynamics are unrepresented in the model, and have not been widely reported as affecting wells drawing from the ESPA. Therefore, we consider the results of our model would be unchanged if seasonal dynamics in aquifer head were more closely aligned with observations.

Simulated irrigation abstractions are generally low compared to USGS observations (Figure S3), and could be increased by either forcing lower efficiency of baseline irrigation practice or increasing evapotranspiration from crops. The efficiency of irrigation technologies is reasonably characterized empirically in the baseline parameterization; however, uncertainties with regard to specific technological parameterizations certainly exist. For instance, the distribution uniformity parameter that controls the amount of water applied to a field during an irrigation event can vary dramatically at field scales (Burt et al., 1997). Following the analysis of Jägermeyr et al. (2015), we use the parameters selected from their sensitivity analysis that optimized trade-offs between crop yield and water use for each technology type. The distribution uniformity, as well as parameters controlling percolation beneath canals and soil infiltration rates, could create less efficient irrigation technologies that would reduce the bias in irrigation water used; however, we avoided calibrating to avoid overfitting with respect to drivers of incidental returns and non-beneficial use. Though unbiased at global scales, the potential evapotranspiration calculation used here (Hamon, 1963) may underestimate the flux from the semi-arid environment of the USRB. Alternatives such as the Penman-Monteith (Monteith, 1965), resulted in poorer representation of the spatial variability in irrigation abstractions though the whole-basin total abstractions were less biased. An increase in irrigation abstractions from higher potential evapotranspiration would increase the baseline CIE by increasing the beneficial consumption of crops, while increasing non-beneficial use and incidental returns only slightly. The excess volume of water lost via simulated discharge from early onset of snowmelt is less than the difference between WBM's and USGS's use estimates for gross irrigation water use in the USRB. Increased abstraction may reduce water available for EAR leading to greater tradeoffs between changing streamflow capture and aquifer drawdown.

### 3.2 Comparison of baseline simulations with other studies

The fraction of incidental returns to the ESPA predicted by simulations is a critical factor for interpreting these results, and we compared simulations with both empirical estimates and previous modelling studies. The fraction of incidental returns in ESPA storage was lower than the fraction of incidental returns entering the aquifer as recharge because the aquifer equilibrates over longer time-scales than the simulations were conducted. The composition of the aquifer was dominated by relict water because we identified all water in the system as relict at the end of spin-up in these simulations to permit tracking fate of all incidental returns. Following sufficient run-time, the model as parameterized at baseline should equilibrate to a composition of at least 60% irrigation returns (Table 2). In 1994 and 1995, isotopic and geochemical tracers showed that water in downgradient portions of the ESPA and in spring outflows consisted of approximately 75% incidental returns from the Snake River (Plummer et al., 2000). The fraction of incidental returns in ESPA recharge was lower in this analysis than estimated from tracers because a) our estimate represents a dilution of incidental returns over the entire ESPA, not local flow-paths sampled near the down-gradient portions of the ESPA where agriculture is concentrated, and b) CIE efficiency from changing irrigation technology decreased rates of incidental recharge between 1994 and 2010 (Dieter et al., 2018; Maupin et al., 2014). These differing assumptions of the amount of irrigation return water in the ESPA is accounted for in our analysis.

The IDWR's ESPAM2.1 apparently predicted greater net recharge from irrigated agriculture to the aquifer; however, direct comparisons are complicated by differing simulation time periods and definitions of simulated fluxes (IDWR, 2013). The

ESPAM2.1 estimates of net recharge accounted for all infiltration from irrigated croplands, whereas we report the infiltration explicitly from applied irrigation water. Net recharge predicted by ESPAM 2.1 was 3.4 km$^3$ y$^{-1}$, greater than the 1.9 km$^3$ y$^{-1}$ of net incidental recharge predicted by WBM at the baseline parameterization. The ESPAM2.1 simulation period was earlier than here (1980 through about 2008), but that model did not exhibit trends in net recharge that would make it consistent with WBM during the later simulation period used here. While the greater groundwater abstractions in the WBM baseline parameterization (2.7 km$^3$ y$^{-1}$) compared to ESPAM2.1 (2.2 km$^3$ y$^{-1}$) may partially explain the difference in net incidental recharge, groundwater abstractions were still lower than the 3.4 km$^3$ y$^{-1}$ estimated by Frans et al. 2012). Both crop type data and meteorological data employed by ESPAM2.1 differ from the data used here (Section 2.2). Wisser et al. (2008) found that combined influence of climate and crop landcover data resulted in uncertainty in crop irrigation demand of up to 50%, consistent with differences between WBM and ESPAM2.1. Considering the low bias in WBM's simulated gross irrigation compared to USGS water-use estimates (Figure S3), and lower rate of net recharge in WBM, we expect that our rates of incidental returns to the system and therefore irrigation reuse are likely underpredicted, at least with respect to the ESPAM2.1. Furthermore, the reduction in net recharge with modernization could be more significant than simulated here, making our estimates of the MB metric potentially low (i.e. conservative).

### 3.3 Baseline simulation water budget and fates

Major fluxes of irrigation abstractions are shown schematically in Figure 2. Beneficial consumption (B) was about 3.52 km$^3$ y$^{-1}$, representing 40% of gross irrigation abstractions (G) at baseline irrigation. Nearly all incidental returns percolated due to the highly permeable geology underlying most of the USRB. About 10% of gross irrigation abstracted, or 0.86 km$^3$ y$^{-1}$ of water (ranging from 0.43 to 1.11 km$^3$ y$^{-1}$), was reused for irrigation each year under the baseline conditions (Figures 1 and 2). Figure 1 shows the spatial intensity of irrigation water reuse (R) for the baseline parameterization. Major controls on the spatial distribution of irrigation reuse included a) administrative basin extent and the balance of incidental returns in reservoirs acting as irrigation source, b) upstream catchment area, and c) presence of the ESPA. Reservoirs received incidental returns in runoff from upstream croplands. The fraction of incidental returns in surface irrigation for entire administrative basins reflect the fraction of irrigation returns stored within the collection of source reservoirs. Therefore, the reuse changed abruptly at administrative basin boundaries (Figure 1). Source reservoirs were not defined in Wyoming at the eastern margin of the model domain. Here, water was provisioned by locating the nearest downstream available water so R increased as incidental returns accumulated along downstream flowpaths. Irrigation reuse changed along the margins of the ESPA as the incidental recharge contributed by groundwater abstractions was characterized by the short-turnover shallow groundwater pool outside the ESPA region. Therefore, in the extreme west of the domain, shallow groundwater contained a high fraction of incidental recharge relative to the ESPA.

Beneficial (BR) and non-beneficial reuse (NR) is calculated explicitly in the model as the beneficial and non-beneficial fraction of gross irrigation reuse. The fraction of beneficial reuse to gross irrigation reuse (BR/R) is roughly equal to basin-wide average classical efficiency (B/G), with slight spatial differences accounting for differing technologies in locations where reuse is more prevalent. Approximately 0.35 km$^3$ y$^{-1}$ of beneficial irrigation consumption is derived from irrigation reuse under our baseline

parameterization (Table 2) representing about 10% of total beneficial consumption (as BR/B), and 8% of total incidental returns (as BR/L).

## 3.4 Effects of irrigation modernization

Modernization of irrigation technology lead to reduced aquifer storage and increased export of water from the basin. Specifically, we find that the modernization decreased plot-scale incidental returns from 4.6 to 0.2 km$^3$ y$^{-1}$ (Figure 3a). As a result, the rate of loss from aquifer storage (drawdown) increases from about 0.7 km$^3$ y$^{-1}$ to about 1.7 km$^3$ y$^{-1}$ when simulated without EAR (Figure 3b), while average annual discharge leaving the basin increases from 10.8 km$^3$ y$^{-1}$ to 12.2 km$^3$ y$^{-1}$ (Figure 3d, Table 2). In these experiments, crop use is independent of irrigation process, so no changes in beneficial crop evapotranspiration are simulated (Figure 3a). Non-beneficial consumption decreased from 0.62 km$^3$ y$^{-1}$ in the baseline to 0.01 km$^3$ y$^{-1}$ for parameterization Eff.I. The high rates of percolation exceeded evaporative demand from bare soils so that incidental recharge was much greater than the non-beneficial consumption from bare soil evaporation.

## 3.5 Enhanced Aquifer Recharge

When simulated EAR ranged from 1.1 km$^3$ y$^{-1}$ (baseline) to 2.4 km$^3$ y$^{-1}$ (Eff.I) to maintain aquifer volume within 0.11 km$^3$ y$^{-1}$ (Figure 3, Table 2). The 120% increase in EAR from baseline to the most efficient parameterization offset a loss of 4.3 km$^3$ y$^{-1}$ from incidental recharge to the aquifer. Incidental recharge from irrigated crops was 4.5 km$^3$ y$^{-1}$ at baseline, and net recharge from irrigated agriculture (incidental recharge minus abstractions) was positive at 1.8 km$^3$ y$^{-1}$ at baseline with or without EAR. Incidental returns represented 60% of water entering the ESPA under baseline conditions (Table 2). As irrigation efficiency increased, incidental recharge to the aquifer decreased, declining to 0.18 km$^3$ y$^{-1}$; incidental recharge flux did not depend on whether EAR was simulated or not. Groundwater abstractions also declined with increasing efficiency; however for the most efficient parameterizations, abstractions exceeded incidental recharge and net recharge from irrigated crops ($I_{rch}$) became negative, declining to -0.8 km$^3$ y$^{-1}$ (Table 2).

We hypothesized that the relative increase in EAR needed to stabilize the aquifer would be less than the loss of net irrigated recharge ($I_{rch}$) resulting from increasing irrigation efficiency. Simulated water balance supported the hypothesis (Figure 4a). For parameterizations more efficient than baseline, the increase in EAR for each parameterization was less than the loss of net irrigated recharge from baseline, and the relationship between the two metrics appeared to be non-linear (Figure 4a). Approximately 72% of the lost net irrigated recharge was required as EAR to stabilize the aquifer for parameterizations Eff.A through Eff.E, and then only approximately 17% of the lost net irrigated recharge was required as EAR for parameterizations Eff.F through Eff.I (Figure 4a). The abrupt change in the effectiveness of EAR to stabilize the aquifer corresponds with an increasing proportion of direct (drip) irrigation for Eff.F through Eff.I (Table 1), reflecting the relatively larger reduction in the distribution uniformity parameter between sprinkler (0.55) and direct (0.05) than from surface (1.15) to sprinkler (0.55), causing rapidly decreased non-beneficial consumption. The magnitude by which the increase in required EAR is less than the relative loss in net irrigated recharge reflects the management benefit (MB) (eq 2) that enhanced aquifer recharge, combined with efficiency, has on aquifer balance (Figure 4a). MB increased to

a maximum of 1.3 km$^3$ y$^{-1}$ for Eff.I , the only parameterization that exceeded the 1.06 km$^3$ y$^{-1}$ EAR needed at baseline to stabilize the aquifer.

For all simulations conducted, the rate of aquifer drawdown (negative $\frac{dV_{ESPA}}{dt}$) was greater than the relative change in flow out of the basin from baseline (Figure 4b). Changing flow out of the basin represents a change in streamflow capture, or how use of water in the basin affects the flux leaving through the river. Simulations with EAR exhibited lower outlet discharge compared to baseline (greater streamflow capture or negative Q*-Q - Figure 4b) as a fraction of Snake River flow was diverted to aquifer replenishment. The rate of EAR controlled the rate of streamflow capture by explicitly adding water to the aquifer, and not through altering the head-dependent baseflow flux back to the river, since increasing EAR also increased baseflow. Note that we focus on changes in streamflow capture relative to baseline, and do not make inference to the absolute fluxes of streamflow capture associated with use of the ESPA. As classical irrigation efficiency increased both with and without EAR, the change in streamflow capture came closer to the rate of aquifer drawdown.

## 4. Discussion

### 4.1 Aquifer reliance on incidental irrigation for recharge

We found a non-linearity in the volumes of enhanced aquifer recharge (EAR) required to stabilize the aquifer as more efficient irrigation technologies were employed. That is, incrementally smaller volumes of enhanced aquifer recharge (EAR) were needed compared to the net irrigated recharge lost due to using more efficient technologies (Figure 4a). This applied only for incremental increases in EAR volumes above requirements for aquifer stability at the baseline parameterization. The volume represented by the efficiency of the combined system from pairing increasing CIE with EAR, the management benefit (MB - Figure 4a) demonstrates that the rate of increasing EAR is less than the rate that net recharge declines. However, the volumetric benefit only exceeded the baseline requirement of EAR for the most efficient (Eff.I) case, and the benefit was not evident for parameterization Eff.A. The management benefit is predominately attributed to additional capture of Snake River discharge (Figures 3 and 4), and by way of increasing irrigation water reuse (Table 2) and decreasing incidental returns (Figure 3). This illustrates that in regions with conjunctively managed surface and groundwater sources like the USRB, increasing basin-wide water resource availability via combined implementation of managed aquifer recharge and changing irrigation efficiency can only be expected to capture more streamflow by transferring water to longer residence time compartments during seasons when water is more available.

Our simulations suggest several important implications of a conjunctive management strategy promoting aquifer recharge while increasing the efficiency of irrigation technology. The amount of EAR needed for a given technology parameterization always exceeded the corresponding rate of drawdown without EAR by 36 to 66%. The excess EAR was needed because the aquifer drainage continues between peak EAR, which follows peak river flow from March to May, and peak irrigation demand (July), and because diverting water away from supply reservoirs shifts reliance to groundwater, which in turn required additional EAR for stabilization. The shift to more groundwater utilization is the primary reason for greater irrigation water reuse for simulations with EAR compared

to simulations without (Table 2).  Furthermore, the rate of aquifer drawdown (up to 1.7 km$^3$ y$^{-1}$ without EAR, and approximately 0 km$^3$ y$^{-1}$ with EAR) more closely approximated changing streamflow capture as CIE increased, meaning that the system converted a greater proportion of the captured streamflow to aquifer storage bringing the change in fluxes into greater parity.  Despite the increasing parity between the rate of drawdown and capture with increasing CIE, drawdown always exceeded the magnitude of the change in streamflow capture (Figure 4b).  This is an expected result because surface water is the dominant source for irrigation and the aquifer is naturally located upgradient of the basin's outlet; interventions that add water to the aquifer (decrease drawdown), will eventually lead to increased downstream discharge (decreased streamflow capture), but the converse is not generally true. Therefore, increasing CIE without EAR will act to deplete the resource relied on by groundwater irrigators more than the impact that EAR would have on downstream users, at least in terms of volumetric shortfalls.

The rate of change in aquifer storage $\left(\frac{dV_{ESPA}}{dt}\right)$ factors significantly in the preceding analysis, and over-estimation of the present-day rate of aquifer drawdown may shift values, but are unlikely to change the general conclusions.  At baseline, we estimated $\frac{dV_{ESPA}}{dt}$ to be -0.71 km$^3$ y$^{-1}$, which is a greater rate of drawdown than the -0.34 km$^3$ y$^{-1}$ estimated by the ESPAM2.1 (IDWR, 2013), the latter being likely more accurate given our underestimates of percolation losses described above.  The rates of EAR we identified to stabilize the lumped representation of the ESPA exceeded both targets and feasible limits of managed aquifer recharge (Idaho Water Resources Board, 2016, Figure 4c).  This potential limitation should be explored in future research focusing on evaluating specific management objectives.

Our simulations assumed a constant beneficial consumption, though use and efficiency are often positively correlated due to the economic incentives to use more water when it is made available locally through efficiency measures (Contor and Taylor, 2013; Grafton et al., 2018; Pfeiffer and Lin, 2014; Tran et al., 2019), and because prior appropriation doctrine requires that water rights holders use their full water right beneficially, essentially encouraging constant levels of water withdrawal regardless of CIE. In the USRB, it is reasonable to assume negligible slippage and rebound effects.  Frequent droughts, the collective action of irrigation districts, and legal agreements between water user organizations outside of the prior appropriation system, all work to incentivize reduced water withdrawals when possible (Gilmore, 2019).  Moreover, a settlement between surface and groundwater irrigators (IDWR, 2015) details specific requirements for ensuring stable aquifer head for both irrigation and downgradient outflow from springs.  To the extent that beneficial use could increase with higher CIE, greater EAR would be required to meet the mandate of aquifer stabilization, or aquifer drawdown would increase without EAR, for any given CIE relative to that simulated here.

The generalizable findings from these simulations have implications for similar semi-arid basins relying on a combination of groundwater and seasonally available surface water. Achieving aquifer stabilization and increasing downstream discharge from combining increased CIE with EAR as simulated here, would require significant investment in hydroinfrastructure of the basin.  In some systems these investments may be a prerequisite for groundwater sufficiency (Scanlon et al., 2016).  In these simulations, EAR was a prerequisite for aquifer stabilization because no tested CIE was able to create a stable aquifer with existing agricultural production and natural recharge alone.  In the USRB, the current head level targeted for stabilization is greater than head existing in the basin prior to irrigation (Kjelstrom, 1995), and generates increased rates of baseflow from springs.  This is a unique issue

from many other semi-arid basins relying on groundwater for irrigation that are managed against aquifer depletion below pre-irrigation heads (Bierkens and Wada, 2019). The primary adverse externality of EAR, aside from technical considerations of feasibility, is decreasing watershed discharge on an annual basis, which would be undesirable for downstream users; however, increasing flow during the irrigation season can be expected (Van Kirk et al., 2020). We found that decreased downstream flow simulated here with EAR, which at 10.2 to 10.6 km$^3$ y$^{-1}$ still exceeds the observed record during the same period (7.35 km$^3$ y$^{-1}$) and existing requirements for instream flow (e.g. 4.1 km$^3$ y$^{-1}$) (IWRB, 1985) by greater than the existing model bias in outlet discharge.

## 4.2 Irrigation Reuse in the Upper Snake River Basin

Incidental returns from irrigation were a major component of the basin's water balance, and therefore are key to understanding basin-scale interpretations of system efficiency. Within the USRB, reuse of incidental returns generated during the model epoch currently makes up at least 9.9% of gross irrigation, and would increase to 14.6% if EAR was used to stabilize the aquifer (Table 2, Figure 2). The baseline value of irrigation water reuse is likely underestimated due to the low bias in gross irrigation, lower rate of net agricultural recharge relative to ESPAM2.1, and a high fraction of relict water composing the ESPA water volume in our simulations. As irrigation efficiency increased and incidental returns decreased both with and without EAR, the total reuse of irrigation water declined (Table 2). However, the fraction of incidental returns that were ultimately used beneficially (beneficial reuse) exhibited very different behaviour if EAR was simulated. With no EAR, beneficial reuse remained between 7 and 8% of total incidental returns for all efficiency parameterizations. With EAR, the beneficial reuse increased steadily with CIE to 30% of total incidental returns for paramterization Eff.I. As a result of the increasing beneficial reuse, basin-scale effective irrigation efficiency either increases faster (with EAR) or slower (without EAR) than classical irrigation efficiency (Figure 5).

Metrics such as the effective irrigation efficiency (EIE) provide a unified metric of efficiency that captures the reusability of incidental returns at the watershed scale (Haie and Keller, 2008). Generally, EIE is calculated using assumptions of the recoverability of irrigation returns; however, we calculate basin-wide EIE using simulated recovered volumes and thereby incorporating explicit estimates of recovered returns. Water within the ESPA was primarily simulated as relict water; therefore the simulations neglect a significant volume of irrigation returns stored within the aquifer from incidental recharge pre-dating the model epoch. Our estimates of irrigation reuse are therefore low, and reflect only reuse of incidental return and subsequent abstractions from the aquifer during the model epoch. We calculate three estimates of EIE, 1) near-term EIE: using the explicitly tracked incidental returns during the model epoch, 2) equilibrium EIE: assuming the equilibrium fraction of incidental returns in abstracted groundwater equals the ratio of incidental to natural recharge (Table 2), and 3) geochemical EIE: assuming that aquifer abstractions consist of a constant fraction of 75% incidental returns estimated geochemically by Plummer et al. (2000). Without enhanced recharge, the actual rate of recovery of incidental returns via irrigation was low, so effective irrigation efficiency is only slightly greater than classical irrigation efficiency at any parameterization (Figure 5), and reflects the large proportion of fresh snowmelt used to supply irrigation most years (Figure 2). With the enhanced recharge, the added reuse increases EIE faster than CIE for parameterizations Eff.A through Eff.G. Assuming equilibrium or geochemical estimates of returns in aquifer water increases estimated EIE by 7 and 11% at baseline, respectively. Moreover, improving irrigation efficiency from baseline through

parameterization Eff.E increases the rate that EIE improves. The parameterizations that correspond with an increasing rate of EIE improvements are the same parameterizations that show a smaller increase in the management benefit, e.g. a smaller amount of additional EAR compensating for the loss of net agricultural recharge (Figure 4a). Though the EIE captures a more complete picture of the effect of changing irrigation technology over the complete system, the high rate of increase in EIE for small changes in irrigation technology may overstate the benefits of intervention on the water balance of the entire basin as captured by the calculation of management benefit.

Incidental returns as a component of discharge at the basin outlet at King Hill, ID was 0.84 km$^3$ y$^{-1}$ (approximately 8% of streamflow) at baseline and declined as CIE increased (Table 2). Therefore, the recoverable incidental returns can be used beyond the USRB. With EAR, incidental returns in discharge were slightly higher than without for each technology parameterization and declined to 0.04 km$^3$ y$^{-1}$ at Eff.I; thereby decreasing incidental returns as a fraction of flow to 0.3%. Modernization acted to increase the unabstracted fraction of discharge leaving the basin, therefore benefitting downstream users while increasing aquifer drawdown. The addition of EAR captured more unabstracted streamflow in the basin, while maintaining a similar flux of exported incidental returns.

Decreasing the fraction of incidental returns in river flow would be expected to improve water quality in the river. However, increasing irrigation reuse implies further recycling of agricultural runoff, which tends towards greater acute water quality threats such as salinization (Ghassemi et al., 1995; Qadir, 2016) and increasing nitrate concentration (Frans et al., 2012). Presently, neither soil salinization nor waterlogging are widespread in the USRB owing to existing conjunctive water abstractions and good drainage, but as irrigation technology modernizes in the USRB and excess irrigation water for flushing is reduced, isolated instances of salinization are becoming increasingly common (Ellsworth, 2004; Moore et al., 2011). While decreasing incidental recharge could exacerbate soil salinization if left unmanaged, irrigation reuse and incidental returns in USRB export both declined (Figure 4, Table 2), which could potentially improve water quality to the ESPA and downstream users. Our definition of incidental returns included canal seepage, a major source of recharge to the ESPA. Canal seepage only represents a source of contaminates if they receive poorly managed runoff, which is not evaluated here. Therefore, in the USRB incidental returns and reuse can only be loosely interpreted as an indicator of water quality, and fate and transport processes would be needed to assess the explicit fate of any agricultural contaminants. Considering the growing concerns of salinization associated with irrigated agriculture (Cañedo-Argüelles et al., 2013; Ghassemi et al., 1995), especially in semi-arid and arid regions with increasing technological efficiencies (Banin and Fish, 1995; Carr G. et al., 2010; Tal, 2016), additional attention is needed to evaluate trade-offs of managing soil salinization and efficiency of irrigation technology.

## 5 Conclusions

Our simulations of the USRB characterize the limitations of relying exclusively on technological adaptation to address water shortfalls in semi-arid regions. Technological modernization does not by itself promote aquifer stabilization in some contexts. Modernization without managed aquifer recharge (MAR) resulted in a greater loss from aquifer storage and increased downstream

flow, undermining the groundwater resource needed for agriculture resiliency in this semi-arid basin. Furthermore, we found that through combined application of MAR and increasingly efficient irrigation technology, the potential increase in downstream flow was always less than the increased drawdown in the aquifer, meaning that less streamflow capture than drawdown was needed for similar crop production in a conjunctively managed system. By increasing MAR to values likely difficult to achieve in practice (IWRB, 2016), the system utilizes only a portion of the net irrigated recharge lost by modernization to stabilize the aquifer. The simulations tested demonstrate the trade-offs inherent in reducing non-consumptive losses through modernization that have been explored in other regions with high gross irrigation reuse (Simons et al., 2015) and illustrate how modernization exports benefits to downstream users. The absence of clear evidence for significantly improved water availability with modernization that is predicted for global scales (e.g. Jägermeyr et al., 2015, 2016; Sauer et al., 2010) is because exported benefits (net increase in water availability) are absorbed by downstream users when analysed at that scale (Grogan et al., 2017). However, in a single headwater semi-arid basin, there is a fundamental lack of parity between local groundwater users and downstream users; any intervention that improves aquifer storage necessarily also benefits downstream users eventually, while the converse is not necessarily true. Also, potential policy and comprehensive water management initiatives which are likely to co-occur with modernization (Gleick et al., 2011) can provide additional benefits to basin water budgets not realized solely by modernizing irrigation technology (Jägermeyr et al., 2015). The ineffectiveness of technological modernization to stabilize the aquifer by itself may reflect the specific setting of the USRB that naturally favours non-consumptive loss to non-beneficial use via high percolation rates coupled with a straightforward avenue for local reuse via a productive aquifer and springs. Irrigation reuse declines as classical irrigation efficiency increases, but using MAR increased the reuse of incidental returns. Though we expected MAR to reduce reuse of incidental returns through the introduction of more pristine water to the aquifer, the larger effect of shifting irrigation reliance towards groundwater from surface water was observed, thereby increasing reuse at the basin scale. The added reuse from implementing MAR in our simulations lead to effective irrigation efficiency increasing faster than classical irrigation efficiency. We would expect the nature of gross irrigation reuse in the USRB to be neither an isolated instance, nor a general exemplar of water allocation issues, but it does provide an example of the complexity and lack of generalizability of specific interventions needed to achieve agricultural sustainability.

## 6 Author Contribution

SZ primarily developed new model code, developed input data, created and ran simulation experiments, created all figures, and drafted text. DG Assisted in development of new model code and input data, and edited text. RL provided guidance on the main research direction, assisted in creation of simulation experiments, and edited text. AP revised new model code, assisted in input data preparation, and edited text. SG assisted in developing input data and provided introductory and background text. PW provided introductory and background text and edited complete draft of text.

## 7 Acknowledgements

This work was supported by the National Science Foundation Innovations at the Nexus of Food, Energy, and Water Systems award 1639524, and Department of Energy Program on Coupled Human and Earth Systems award DE-SC0016162. The authors acknowledge the technical support of S.Glidden, and the constructive feedback provided by J.M. Davis on early drafts of the manuscript.

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

**Figures**

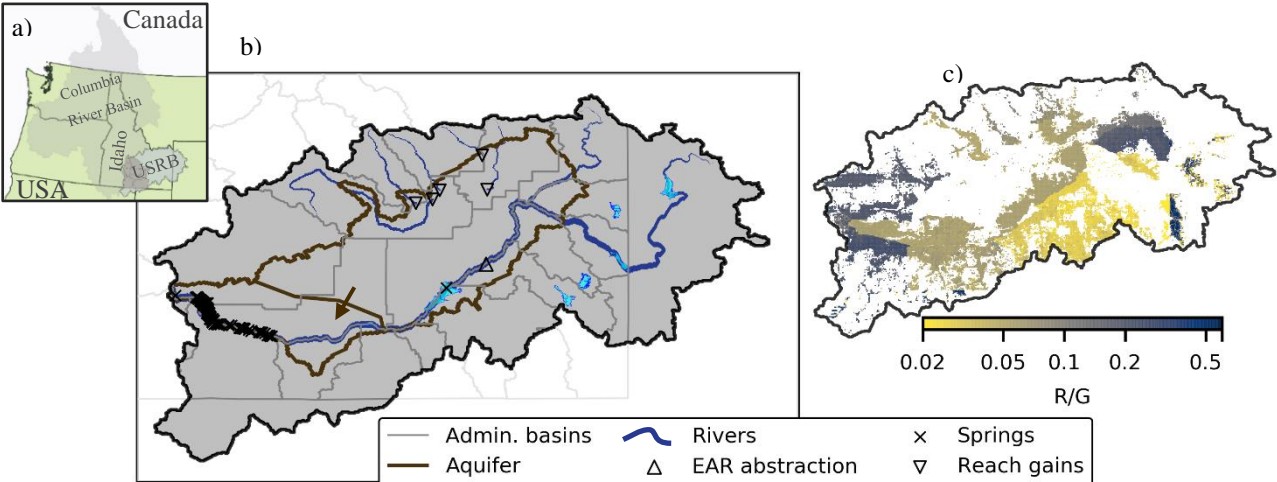

**Figure 1: Location of Upper Snake River Basin (USRB), a headwaters of the Columbia River, in the US States of Idaho, and Wyoming (a). Configuration of major hydrologic features of USRB, including the two compartments conceptualized for the Eastern Snake Plain Aquifer (ESPA), locations of reach gains where losing rivers drain to the ESPA, the location of simulated abstractions for enhanced aquifer recharge (EAR), springs where flow from the ESPA drains back to the Upper Snake River, the river network scaled by mean annual flow, and extents of administrative basins (IDWR, 2015) indicating areas using common surface water sources for irrigation (b). Average fraction of gross irrigation comprised of incidental returns (irrigation reuse - *R*) (c).**

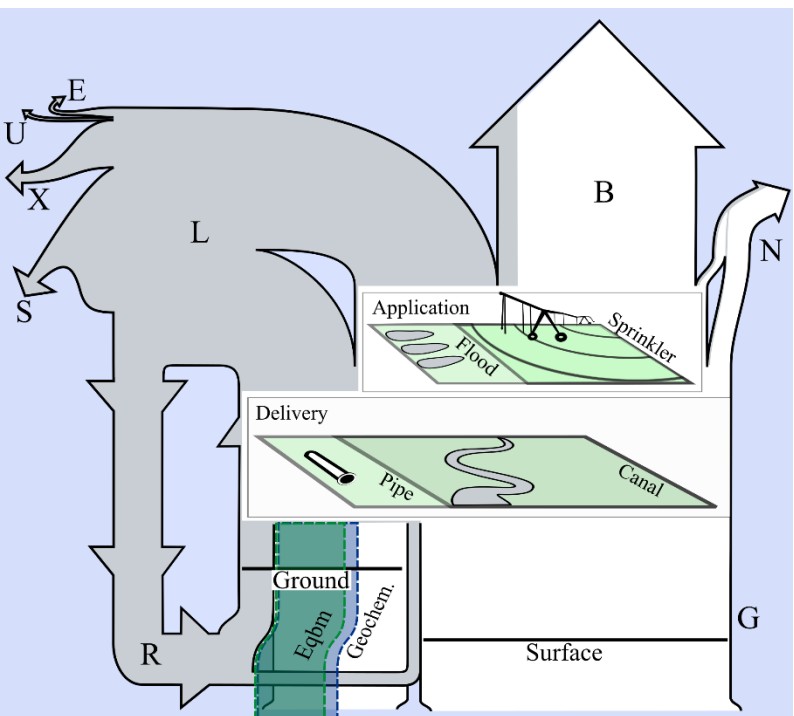

**Figure 2:** **Diagram of fates of water abstracted for irrigation across the USRB. Flow-line widths are scaled proportional to fluxes across the simulation domain between 2008 and 2017 at the baseline parameterization. White depicts abstractions from pristine sources, whereas water lost non-consumptively from irrigation delivery or application during the model epoch is grey. Equilibrium (Eqbm) and geochemical (Geochem.) fractions of groundwater abstractions relax assumptions about aquifer water composition and are discussed in Section 4.2. Labels of irrigation fluxes are: G – gross irrigation abstractions, B – beneficial consumption by crops, N – non-beneficial consumption, L – non-consumptive losses or incidental returns, and the remaining fluxes refer to the fate of incidental returns: R – reuse in gross irrigation, E – evaporation, U – human use, X – export, and S – storage in aquifer, soils, and reservoirs.**

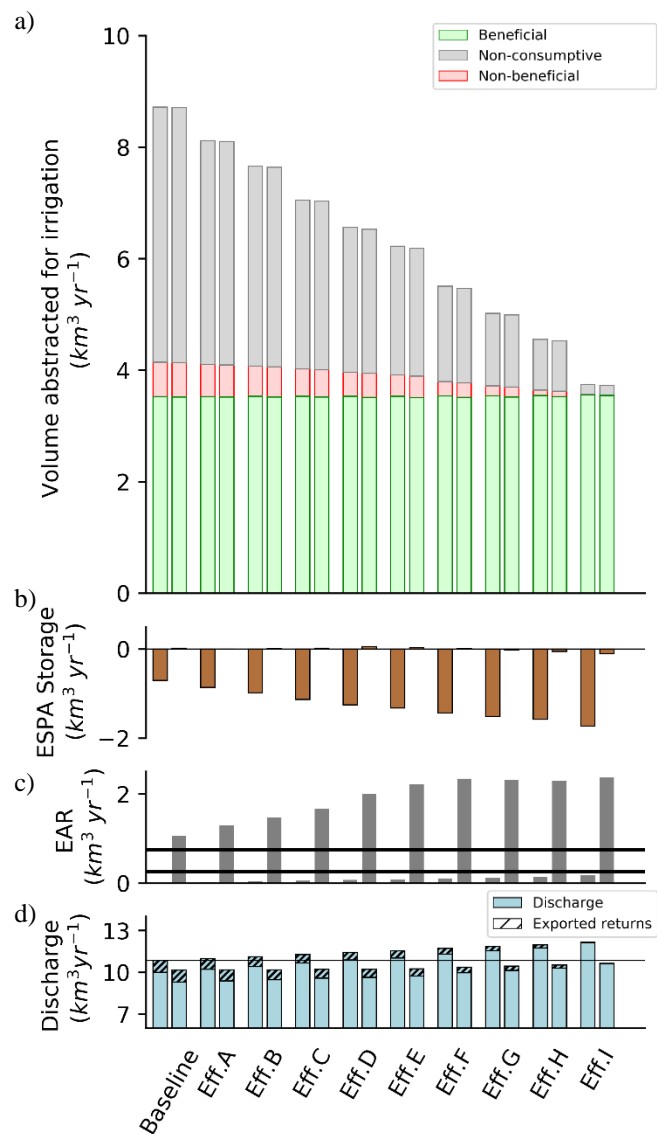

**Figure 3: (a)** Critical water fluxes across efficiency scenarios paired by simulations without (left) and with (right) enhanced aquifer recharge (EAR). Component fractions of gross irrigation water for the USRB as 2008-2017 averages. **(b)** Average change in volume of the ESPA. **(c)** Enhanced aquifer recharge (recharge to the ESPA upstream of American Falls) required to stabilize the aquifer water balance. Horizontal lines represent target (0.26 km³ y⁻¹) and feasible (0.75 km³ y⁻¹) bounds on existing managed aquifer recharge practice and infrastructure (IWRB, 2016). **(d)** Discharge and exported incidental returns at the watershed outlet at King Hill, Idaho. Horizontal line indicates average discharge at baseline.

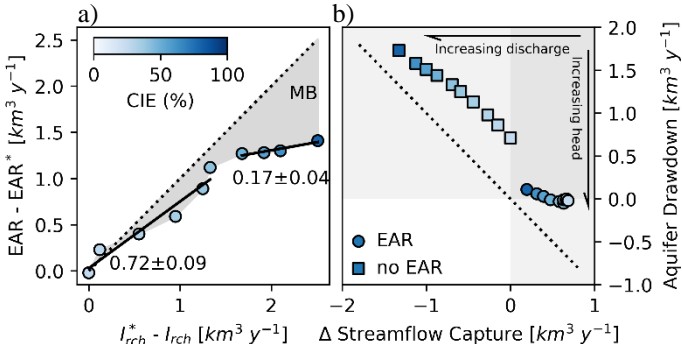

**Figure 4: (a)** Enhanced aquifer recharge (EAR) above the EAR required at baseline (EAR*) to ensure aquifer stabilization plotted against the reduction in net recharge from irrigation ($I_{rch}$) from baseline ($I_{rch}^*$) as classical irrigation efficiency (CIE) increases. Dotted line represents equal increases in EAR and reductions in net recharge. All scenarios show that less additional EAR is required than is lost from net recharge as CIE increases, the magnitude is referred to as the management benefit (MB). Slopes of piecewise linear regressions (black lines) between two variables are shown with standard error of the estimate. **(b)** Aquifer drawdown plotted against change in basin streamflow capture (Q*- Q) with dotted line representing equal changes to discharge from baseline and drawdown.

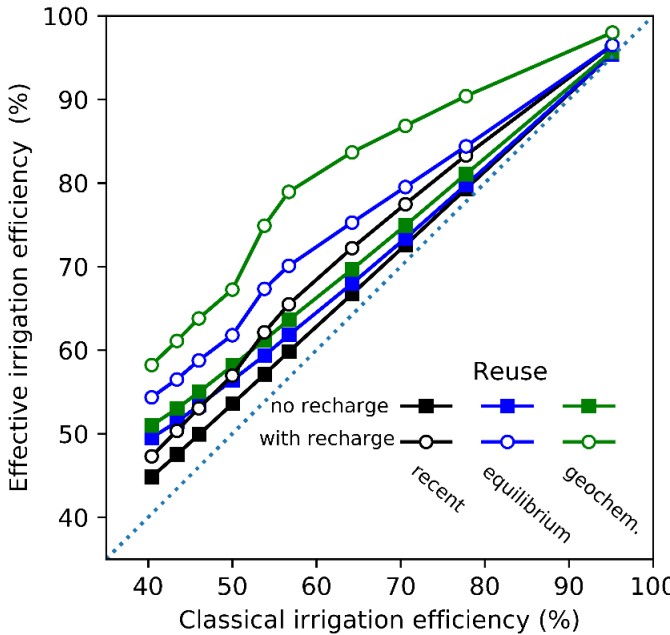

**Figure 5: Effective irrigation efficiency plotted against the classical irrigation efficiency of each parameterization. Effective irrigation efficiency is calculated three ways based on the estimates of irrigation reuse: near-term – simulated reuse where incidental returns in aquifer abstractions is represented explicitly during the model epoch (3%), equilibrium – incidental returns in the aquifer abstractions are assumed to be at equilibrium in the aquifer at ratio of incidental recharge to total recharge (Table 2), and geochemical – incidental returns in aquifer abstractions are assumed to be represented by an average estimated geochemically (Plummer et al., 2000) (75%).**

## Tables

**Table 1: Definition of efficiency parameterizations.**

| | | Fraction of | | |
| Parameterization | baseline surface irrigation | open-surface conveyances | Unlined canals | Fraction of direct irrigation |
|---|---|---|---|---|
| Baseline | 1.0 | 0.83 | 1.0 | 0.00 |
| Eff.A | 0.89 | 0.82 | 0.95 | 0.10 |
| Eff.B | 0.86 | 0.80 | 0.8 | 0.10 |
| Eff.C | 0.82 | 0.70 | 0.60 | 0.10 |
| Eff.D | 0.77 | 0.60 | 0.40 | 0.10 |
| Eff.E | 0.73 | 0.50 | 0.20 | 0.10 |
| Eff.F | 0.53 | 0.40 | 0.15 | 0.33 |
| Eff.G | 0.40 | 0.30 | 0.10 | 0.50 |
| Eff.H | 0.27 | 0.15 | 0.05 | 0.67 |
| Eff.I | 0.0 | 0.0 | 0.0 | 1.00 |

Table 2: Comparison of irrigation related water fluxes in and out of the Eastern Snake Plain Aquifer and Upper Snake River Basin along a gradient of increasing efficiency of irrigation technology.  Values without and with enhanced aquifer recharge depicted on left and right halves of each column, respectively.

| | Baseline | | Eff. A | | Eff.B | | Eff.C | | Eff.D | | Eff.E | | Eff.F | | Eff.G | | Eff.H | | Eff.I | |
|---|---|---|---|---|---|---|---|---|---|---|---|---|---|---|---|---|---|---|---|---|
| | | EAR | | EAR | | EAR | | EAR | | EAR | | EAR | | EAR | | EAR | | EAR | | EAR |
| Classical irrigation efficiency (%) | 40.4 | 40.4 | 43.5 | 43.4 | 46.1 | 46.0 | 50.1 | 50.0 | 53.8 | 53.8 | 56.8 | 56.7 | 64.2 | 64.2 | 70.6 | 70.6 | 77.8 | 77.8 | 95.1 | 95.2 |
| Incidental recharge (fraction) | 60 | 53 | 56 | 48 | 54 | 45 | 49 | 40 | 44 | 36 | 42 | 31 | 35 | 25 | 29 | 20 | 23 | 15 | 5 | 3 |
| Groundw. abstraction (km$^3$ y$^{-1}$) | 2.64 | 2.88 | 2.39 | 2.31 | 2.32 | 2.39 | 2.12 | 2.26 | 1.91 | 2.23 | 1.83 | 1.9 | 1.57 | 1.7 | 1.4 | 1.55 | 1.28 | 1.33 | 0.95 | 1.02 |
| Net irrigated recharge ($I_{rch}$ km$^3$ y$^{-1}$) | 1.83 | 1.67 | 1.52 | 1.55 | 1.25 | 1.12 | 0.86 | 0.72 | 0.65 | 0.42 | 0.42 | 0.34 | 0.12 | -0.01 | -0.12 | -0.25 | -0.36 | -0.43 | -0.77 | -0.84 |
| Irrigation reuse (R) (km$^3$ y$^{-1}$) | 0.86 | 1.27 | 0.7 | 1.12 | 0.59 | 1.01 | 0.47 | 0.86 | 0.38 | 0.87 | 0.32 | 0.83 | 0.2 | 0.61 | 0.14 | 0.45 | 0.09 | 0.3 | 0.01 | 0.05 |
| Beneficial Reuse (BR/R) (%) | 7.67 | 11.3 | 7.57 | 12.5 | 7.65 | 13.1 | 7.73 | 14.7 | 7.81 | 18.9 | 7.86 | 21.3 | 7.64 | 23.8 | 7.55 | 25.2 | 7.64 | 26.9 | 6.85 | 29.7 |

*EAR* – Enhanced aquifer recharge
*USRB* – Upper Snake River Basin

