# Peer review of "Interplay of changing irrigation technologies and water reuse: Example from the Upper Snake River Basin, Idaho, USA"

_Hydrology and Earth System Sciences, 2020_

## Referee Comment (RC1) · Anonymous Referee #1 · 9 May 2020

Review of manuscript HESS-2020-135, entitled "Interplay of changing irrigation technologies and water reuse: Example from the Upper Snake River Basin, Idaho, USA" by S. Zuidema, D. Grogan, A. Prusevich, R. Lammers, S. Gilmore, and P. Williams This paper describes a study where a distributed hydrological model was used to investigate the effects of managed aquifer recharge (MAR) on the system-scale efficiency of modernizing irrigation technology and the resulting changes to the reuse of non-consumptive losses in the semi-arid Upper Snake River Basin (USRB) of western Wyoming and southern Idaho, USA. The manuscript is well written, organized, and certainly fits the scope of a journal like HESS. Authors proper acknowledge the difficulties in modelling a complex system such as the Upper Snake River Basin, with reservoirs

management influencing river flow, and irrigation management the aquifer recharge, by taking into account modelling uncertainties and bias when discussing the main findings extracted from their simulation scenarios. In my opinion, the paper should be accepted for publication in HESS after some minor review. Minor comments: Page 1, L3: There seems to be a misplaced "and" in the middle of authors names. Page 2, L11-13: Please elaborate more on this statement. It is not very clear why the economics of running a more efficient system leads to an increased water consumption. Page 5, L3: Please check the need of including L twice in the text between brackets. Page 5, L4: Please check if infiltration or percolation. P4, L31: Please explain clearer how irrigation efficiency was defined. Was it in terms of factions of B + N + L? Page 7, L2: It should be soil water content above field capacity, not saturation. Page 7, L14-18: If I understood it correctly, you defined consumptive and non-beneficial losses based on the characteristics of irrigation methods. Can you add some examples or assumptions made? Page 8, L4: Did reservoir data include consumptions also? Page 8, L22: What was the adaptation made to FAO56? Page 8, L22-23. There is no such thing as reference PET. There is a reference ET, a crop ET (that refers to crop evapotranspiration potential values) and an actual ET. Please check Allen et al. (1998). Page 9, L2-8: Are these comparisons between observations and model outputs (flow, river discharge, storage) on a daily or monthly basis? Page 9, L24-27: I agree that hydrological modelling of basins with reservoir is a challenging task. This is even more true when using distributed models. I just don't see why you choose to cite a paper still under review when there is already some literature on this issue. Page 10, L16-18: Beneficial consumption of water refers to plant transpiration while non-beneficial consumption is the soil evaporation component. In your model how is ET partitioned in these two components? Unless I've missed this was not explained.
* * *

---

## Referee Comment (RC2) · Anonymous Referee #2 · 24 May 2020

The paper explores the impacts of conjunctive introduction of irrigation modernization and managed aquifer recharge in a semi-arid setting, by analyzing different scenarios with a physically-based simulation model. With its conceptual approach and presentation, the paper succeeds in providing new insights in the functioning and management of such a complex system, which makes it highly relevant to the scientific community. The paper is very well written and structured and certainly fits the scope of HESS. My opinion is that it should be accepted after minor review.

Specific comments:

The paper acknowledges that CIE changes can lead to increased consumption at the

basin scale. This phenomenon mainly occurs when no physical or legal limitations are in place for irrigators to expand their operations (e.g. by increasing surface area or switching to more water-intensive crop types) with the "saved" water. The simulations described in lines 21-25 on page 3 seem to disregard this effect, which is in practice so often observed. It may be outside the scope of the paper to implement this behavioral aspect in the simulations, but it is important to include at least in the discussion. Are the farmers in the USRB in some way incentivized to reduce their withdrawals as CIE increases? If not, it should be acknowledged that this makes the simulations more theoretical than realistic. If yes, this is still relevant for more elaborate discussion, as results are presented in a generalized context of semi-arid basins, which for the large majority do not have such restrictions effectively in place.

Section 3.1 acknowledges model validation issues and discusses implications for the findings of the study. It is quite positive that this is so transparently discussed and helps to put the results into perspective. In particular, snowmelt processes and underestimated irrigation demand issues are discussed here. There are two other striking observations in the model validation demonstrated in the supplementary material, namely the negative NSE in reservoir storage and the lack of representation of seasonal dynamics (at least, in the springs time series). Supposedly, these would affect water availability for irrigation and partitioning in the surface vs. ground water use (or not?). To me it would seem that implications of these two model quality issues can be discussed more explicitly.

The distribution uniformity parameter assumed for different irrigation types seems to have a large impact on the effectiveness of EAR to stabilize the aquifer, as described on page 13 (lines 1-2). This relates to a key result from the study. It would be of added value to provide more information on the way these values were determined (in the Jaegermeyr study), what are uncertainties associated with them, and how these would affect the outcomes of the paper.

Page 15 introduces the Effective Irrigation Efficiency (EIE). It would be good to define

EIE somewhere in the paper using the terms of B, L, etc., to allow for better understanding of the discussion in Section 4.2 which relates EIE to CIE. To my understanding, the key distinguishing feature of EIE is that it includes water quality in its definition, which is not explicitly modeled in this study.

Technical corrections:

Page 4, Line 26: " provide irrigate"

Figure 1b: the representation of the black shape in the west of the catchment (a line of X's?) can be improved.

Page 12, Line 17: needs rephrasing

---

## Referee Comment (RC3) · Anonymous Referee #3 · 24 May 2020

This paper describes a simulation study using a distributed hydrological model with adjustments to irrigation efficiency parameters and calculation of managed aquifer recharge required to sustain groundwater levels. I must first disagree with the other two reviewers; I found the methodology very difficult to read and understand. More importantly, I struggled to grasp the point of the analysis and its contribution to water management. The hypothesis is presented (in the abstract) as follows: "as efficiency improves, less MAR is required to maintain a stable aquifer than returns flows are reduced due to increased efficiency". Inefficient irrigation has the unintended benefit of providing water to downstream users. So efficiency improvements result in a loss of that water provision, which can be offset by MAR. Since the net effect of efficient ir-

rigation and MAR would be to reduce evaporative losses, I cannot see why there is any surprise in the result attained in this study. The conclusion seems to follow directly from an assumption of the modeling (although I can see from the other reviews provided that I may lack some important understanding here). My main recommendation is that the authors work on the introduction to explain clearly why this very detailed modeling effort is needed to reach what is seemingly a very obvious fact that could be deduced with some basic logic.

---

## Author Comment (AC1) · 9 Jul 2020

We thank the referee for their generous review of the work, and the constructive feedback provided. We agree with the comments and will address each in turn. In the revision of the manuscript, we will easily fix the confusing word choice or grammatical errors identified by the referee at: (P1, L3), (P5, L3), (P4,L4 [percolation]), and (P7, L2). We would like to respond to more substantive comments directly here.

We make passing comment (P2,L11) to the economic pressure to utilize more water, when that water is abstracted more efficiently. Contor and Taylor (2013) provided an illustrative example of this paradox in the context of the Upper Snake River Basin.

We will more clearly capture the essence of their thesis in the revised manuscript. Basically, a more efficient irrigation system reduces non-consumptive water losses, thereby making more water available for consumptive use. In a water rights system where an irrigator has a fixed amount of water available to them and a mandate to use all of their water beneficially, reductions in non-consumptive water losses are likely to be balanced by an expansion of cropland or planting of crop with greater ET to make use of the "extra" water. This phenomenon is well known within the resource economics literature, where it is referred to as "Jevon's Paradox". Our revised manuscript will streamline this summary in our introduction.

In introducing our definitions of hydrologic fractions and irrigation efficiency (P4,L31), the referee points out that we did not provide a specific definition of irrigation efficiency. Though this was intentional as we wanted to focus our use of the term "irrigation efficiency" to the discussion, upon reflection we agree that it is unclear as written. We describe both classical irrigation efficiency (CIE) and effective irrigation efficiency (P15,L13) so we agree the manuscript is improved by clearly defining these early. We will provide the definition of CIE:

CIE=B/G

and an explicit equation for EIE, (not provided in the original manuscript, which was an oversight):

EIE=B/(G-R)

when we introduce hydrologic fractions. In the above equations, B, G, and R refer to beneficial consumption, gross irrigation abstractions, and reused irrigation abstractions, respectively, all in units of volume per time.

The referee requests additional clarity regarding the calculation of "consumptive and non-beneficial losses" (which will be revised to more accurately state "non-beneficial consumption") at P7, L14. Though the details of this calculation are provided in the

supplement, we agree that examples defining the processes involved are warranted. For example a proposed revision (P7, L14):

"The system explicitly represented non-beneficial consumption as evaporation of sprinkler mists and evaporation from canal and soil surfaces, using technology specific parameters reflecting county-wide averages from USGS water use statistics (Dieter et al., 2018; Maupin et al., 2014). A representative fraction of 4% of sprinkler applied water is evaporated as mists (Bavi et al., 2009; McLean et al., 2000; Uddin et al., 2010). Further, during the irrigation season, water is assumed to be evaporating at potential rates throughout the canal network. We assume crop ET is required (e.g. beneficial) for both transpiration and salt flushing, but water applied during an irrigation event in excess of daily crop demand wets soil above field capacity. This water evaporates (non-beneficially) at the potential rate, and unevaporated water is returned non-consumptively at the end of the timestep via either percolation, or runoff if vertical hydraulic conductivity is too low. The algorithm describing irrigation water fates is detailed in the Supplemental material."

Reservoir data retrieved from the US Bureau of Reclamation did not include direct estimates of consumption from reservoirs (P8,L4) by either non-beneficial evaporative consumption or gross abstractions. Although we do calculate surface evaporation from reservoirs, we do not associate this flux with non-beneficial consumption as water is stored for multiple uses in all reservoirs making attribution of open-water evaporation to a particular sector (such as irrigated agriculture) problematic.

Our word choice at the opening of the paragraph referring to WBM's use of the FAO56 (P8,L22) is confusing and will be revised. It is more accurate to say that "WBM uses" or "WBM adapted" FAO56 (Allen et al., 1998) to estimate crop water demands. Although we utilize alternative techniques for calculating reference ET (e.g. Hamon (1963) here), crop water demands follow directly from FAO56. This will be clarified in our revisions.

We thank the referee for catching our misuse of terminology in describing reference ET

on (P8,L22).

In the revised manuscript we will clarify the sample duration associated with each of the four metrics used to establish model performance at P9, L2-8. This information is summarized in Table S2: monthly discharge from springs, annual gross irrigation by county, seasonal mean discharge, and seasonal mean reservoir storage.

Indeed there is a breadth of knowledge (e.g. Adam et al., 2007; Masaki et al., 2017) regarding the challenges of representing reservoir operations that we will cite (at P9,L24) in the revised manuscript. We drew from (Rougé et al., In revision) because it deals specifically with issues of using the same simulation model in the same domain, and we are confident of a successful review prior to publication of our submitted piece.

Details regarding the calculation of the partition between beneficial and non-beneficial are detailed in the manuscript's supplement. Considering the referee's comment here (P10,L16) and previously (P7,L14), we are adding a bit more detail to the body of the manuscript (presented above) and a stronger call to review the supplement for more information.

References:

Adam, J. C., Haddeland, I., Su, F. and Lettenmaier, D. P.: Simulation of reservoir influences on annual and seasonal streamflow changes for the Lena, Yenisei, and Ob' rivers, J. Geophys. Res. Atmospheres, 112(D24), doi:10.1029/2007JD008525, 2007.

Allen, R. G., Pereira, L. S., Raes, D., Smith, M. and others: Crop evapotranspiration-Guidelines for computing crop water requirements-FAO Irrigation and drainage paper 56, FAO Rome, 300, 1998.

Bavi, A., Kashuli, H. A., Boroomand, S., Naseri, A. and Albaji, M.: Evaporation Losses from Sprinkler Irrigation Systems under Various Operating Conditions, J. Appl. Sci., 9(3), doi:10.3923/jas.2009.597.600, 2009.

Contor, B. A. and Taylor, R. G.: Why improving irrigation efficiency increases total volume of consumptive use: irrigation efficiency increases consumptive use, Irrig. Drain., 62(3), 273–280, doi:10.1002/ird.1717, 2013.

Dieter, C. A., Maupin, M. A., Caldwell, R. R., Harris, M. A., Ivahnenko, T. I., Lovelace, J. K., Barber, N. L. and Linsey, K. S.: Estimated use of water in the United States in 2015, USGS Numbered Series, U.S. Geological Survey, Reston, VA. [online] Available from: http://pubs.er.usgs.gov/publication/cir1441 (Accessed 13 November 2018), 2018.

Hamon, W. R.: Computation of direct runoff amounts from storm rainfall, in International Association of Hydrological Sciences Publications, vol. 63, pp. 52–62, International Association of Hydrological Sciences., 1963.

Masaki, Y., Hanasaki, N., Biemans, H., Schmied, H. M., Tang, Q., Wada, Y., Gosling, S. N., Takahashi, K. and Hijioka, Y.: Intercomparison of global river discharge simulations focusing on dam operation—multiple models analysis in two case-study river basins, Missouri–Mississippi and Green–Colorado, Environ. Res. Lett., 12(5), 055002, doi:10.1088/1748-9326/aa57a8, 2017.

Maupin, M. A., Kenny, J. F., Hutson, S. S., Lovelace, J. K., Barber, N. L. and Linsey, K. S.: Estimated use of water in the United States in 2010, U.S. Geological Survey, Reston, Virginia., 2014.

McLean, R. K., Ranjan, R. S. and Klassen, G.: Spray evaporation losses from sprinkler irrigation systems, Can. Agric. Eng., 42(1), 8, 2000.

Rougé, C., Reed, P., Grogan, D., Zuidema, S., Prusevich, A., Glidden, S., Lamontagne, J. and Lammers, R.: Coordination and Control: Limits in Standard Representations of Multi-Reservoir Operations in Hydrological Modeling, Hydrol Earth Syst Sci, doi:https://doi.org/10.5194/hess-2019-589, In revision.

Uddin, J., Smith, R., Hancock, N. and Foley, J. P.: Droplet evaporation losses during sprinkler irrigation: an overview, in Australian Irrigation Conference and Exibition 2010: Proceedings, edited by K. Montagu, pp. 1–10, Irrigation Australia Ltd., Sydney, Australia. [online] Available from: http://www.irrigationaustralia.com.au/visitor.cfm?id=41 (Accessed 2 April 2019), 2010.

---

## Author Comment (AC2) · 9 Jul 2020

We would like to start by thanking the referee for their thoughtful and generous comments on our manuscript. We found the feedback constructive and discuss how these comments will inform the revisions of our manuscript.

The referee astutely recognized that we did not include a representation of Jevon's paradox in the context of irrigation efficiency, even though we highlight how it has affected water management in the USRB in the past (P2,L11). Though we don't believe that it is feasible for us to make such a representation, it is important to discuss the effect of a positive correlation between consumptive use and efficiency and plan to

provide such a discussion in the final paragraph of section 4.1 (P14,L21). There are two competing incentives at work here with regards to water withdrawal reductions as CIE increases: on the one hand, prior appropriation doctrine requires that water rights holders use their full water right beneficially, essentially encouraging constant levels of water withdrawal regardless of CIE. On the other hand, frequent droughts, the collective action of irrigation districts, and legal agreements between water user organizations outside of the prior appropriation system, all work to incentivize reduced water withdrawals when possible (Gilmore, 2019). Moreover, a settlement between surface and groundwater irrigators (IDWR, 2015) details specific requirements for ensuring stable aquifer head for both irrigation and downgradient outflow from springs. For the simulations shown here, a reduction in water withdrawals as CIE increases is not completely unrealistic due to the documented, incentivized, and coordinated effort underway in the USRB to stabilize aquifer levels.

We would be happy to accommodate the referee's comment regarding a discussion of seasonality of discharge from springs and reservoir storage. Our revised manuscript would more clearly relate the timing of snowmelt and release through the cascade of reservoirs discussed at P10,L4-10 as the reason for reservoir volume misfit. The seasonal dynamics of spring outflows are highly damped in the model because of the lumped approach by which we simulate groundwater storage, and this deserves mention. Though we understand the mechanisms that control this misfit, we expect a thorough explanation of these will be distracting in the context of the manuscript. Still, we agree that it prudent and appropriate to state (in the results section) that a) seasonality does not affect internal functioning of groundwater abstraction, and there is no evidence that seasonal variation in the water table creates widescale problems for groundwater pumpers, and b) springs discharge to a point on the Snake River downstream of major surface water abstractions in the basin, so seasonality of discharge from springs does not affect surface water irrigators in the USRB. Our consideration of downstream flow and availability to downstream irrigators only focuses long-term effects of changing flow at annual time-scales.

[Figure]

We agree that the Distribution Uniformity (DU) parameter, which controls the excess water each technology applies to fields, is critical in our study and will expand on the definition (e.g. Burt et al. 1997) and use of the concept in our revised manuscript. The DU parameter controls the amount of non-consumptive returns from irrigation, which then determine the amount of enhanced aquifer recharge (EAR) required for aquifer stabilization. Therefore, the referee is correct to point out that uncertainty in these values may influence the quantitative results of our study. We note that the work of Jägermeyr et al. (2015) show that both crop yields and soil moisture deficit are fairly insensitive to a range of DU in the regions selected such that modest reductions in DU (increases in efficiency) would have little adverse impact on cropping outcomes, but increases in DU (decreasing efficiency) would incur no benefits. We agree that this is a valid point, but we are unclear what to do about this uncertainty. Consider two cases. Case 1, DU are selected such that they are lower for sprinkler and surface irrigation so that crop beneficial consumption is virtually unchanged, however incidental recharge is reduced so that EAR must therefore increase to ensure aquifer stabilization. This may be a practical scenario for system management; however, it is difficult to reconcile with observations from the USRB because our model already is biased low for gross irrigation abstractions and such a parameterization would deviate further from observations. Now for Case 2, the DU parameters could be increased such that the system is less efficient and increasing incidental recharge offsets a need for a certain amount of EAR. Although this would better fit the high rates of gross irrigation abstraction, it does not seem likely that a heavily regulated and expensive water distribution system would permit for such inefficiencies. Furthermore, such a parameterization without an empirical basis (as provided by the analysis of Jägermeyr et al. 2015) would be overfitting the model to the limited observational data we have. Our final point on this topic is a general one. The DU parameter is but one of several very uncertain parameters that have an identical effect on the system outcome including vertical saturated hydraulic conductivity underlying canals, the existing quality (anecdotally poor) of canal liners, the proportion of active irrigation technologies, and infiltration rates of soil. Uncertainty

in these parameters likely overwhelms that of the DU and a substantial portion of the range of the cumulative effect is characterized by our scenarios. Still the importance of this parameter, and the capacity to improve a number of factors associated with irrigated water associated with improved water management (practices that would actually reduce the DU) will be more clearly identified in the conclusion of the revised manuscript.

Yes indeed, another reviewer also identified our need to define effective irrigation efficiency (EIE) in the manuscript which was an oversight. While Haie and Keller (2008) define EIE using a water quality discount in certain classes of models, they also define a quantity only case that considers all incidental returns as available for further use so that the denominator used in the calculation of EIE reflects only the blue water abstracted for irrigation:

EIE=B/(G-R)

where B, G, and R represent beneficial consumption, gross irrigation abstraction, and reused irrigation abstraction, respectively, all in units of volume per time. This equation and the equation for classical irrigation efficiency will be included in the discussion of hydrologic fractions in Section 2.2 of the revised manuscript.

We thank the referee for pointing out the needed technical corrections.

References:

C. M. Burt, A. J. Clemmens, T. S. Strelkoff, K. H. Solomon, R. D. Bliesner, L. A. Hardy, T. A. Howell and D. E. Eisenhauer: Irrigation Performance Measures: Efficiency and Uniformity, J. Irrig. Drain. Eng., 123(6), 423–442, doi:10.1061/(ASCE)0733-9437(1997)123:6(423), 1997.

Gilmore, S.: Assessing the Adaptive Capacity of Idaho's Magic Valley As a Complex Social-Ecological System, MS, University of Idaho, Moscow, ID., 2019.

Haie, N. and Keller, A. A.: Effective Efficiency as a Tool for Sustainable Water Resources Management, JAWRA J. Am. Water Resour. Assoc., 44(4), 961–968, doi:10.1111/j.1752-1688.2008.00194.x, 2008.

IDWR: Settlement Agreement Entered into June 30, 2015 Between Participating Members of the Surface Water Coalition and Participating Members of the Idaho Ground Water Appropriators, Inc. [online] Available from: https://idwr.idaho.gov/files/legal/swc-igwa-settlement/SWC-IGWA-Settlement-20150630-SWC-IGWA-Settlement-Agreement.pdf (Accessed 1 July 2020), 2015.

Jägermeyr, J., Gerten, D., Heinke, J., Schaphoff, S., Kummu, M. and Lucht, W.: Water savings potentials of irrigation systems: global simulation of processes and linkages, Hydrol Earth Syst Sci, 19(7), 3073–3091, doi:10.5194/hess-19-3073-2015, 2015.

---

## Author Comment (AC3) · 9 Jul 2020

We thank the referee for their comments. We will try to address the concerns here and highlight areas where we think we can make the work more understandable for readers as highlighted by the review. Other reviewers made minor points that can help clarify the methodology section, and we will attempt to clarify the description regarding the experiment structure (2.3), which we thought was the most challenging to convey.

We consider that the referees second point in a positive light because good science tends to be unsurprising in retrospect. The hypothesis the referee refers to is described in its entirety at the end of the introduction (P3,L18-21) and repeated here:

[Figure]

"We hypothesized that only a fraction of the reduced incidental returns from modernizing technology would be needed to maintain aquifer volume if introduced as MAR. An alternative hypothesis is that asynchronicity in recharge water availability and irrigation demand would require greater recharge rates than if water was introduced as inefficient irrigation and reused contemporaneously."

We acknowledge an alternative hypothesis because the conclusion is not forgone as this is a complex system with interfering processes (e.g. increasing efficiency both reduces recharge to the aquifer, but also reduces pumping from the aquifer). As a result of these processes, the hypothesis is supported, but in (what we consider) and interesting and non-linear way (P13,L18-20).

One of the benefits of modeling any complex system is to elucidate non-linear emergent behavior. Here, the results with regards to MAR volumes versus return flow reductions is clearly non-linear. Each aspect of the model is represented using straightforward and first-order assumptions, yet there is no clear way (to us) to evaluate which processes supersede others in a given context except to encode these assumptions into a distributed simulation model of the system. When we ascertain that the rate of reduction in non-beneficial losses from increasing irrigation efficiency outpaces the increasing rate of managed aquifer recharge, we have certainly identified a first-order process that is responsible for that outcome. But we have established that finding in concert with numerous other first-order processes that may have lead to different outcomes in a different context. This type of modeling study adds to an ongoing discussion in the literature – as cited in the introduction and discussion sections – on what metrics of efficiency and re-use are informative at the watershed management scale. Long-used metrics such as classical irrigation efficiency (CIE) which are field-scale and useful for the engineering of individual irrigation systems, and even the effective irrigation efficiency (EIE), which accounts for the reusability of return flows, fall short of this goal. Hydrologists are seeking new ways to quantify and evaluate irrigation management at watershed scales. Here, we see that certain characteristics of the watershed such as the high percolation rates, tight connection between surface water and groundwater, and the existing regulatory framework, are all important considerations when constructing an informative efficiency metric targeted at achieving local water management goals.

The referee's final suggestion is an important one, which we expect will greatly improve the manuscript. Though we allude to the robust discussion of the importance of assessing the interplay between reuse of non-consumptive losses and improving efficiency in our introduction, we can improve it to give more context describing the utility of model experimentation, and alternative methodology such as utilizing natural experiments where alternative practices are incentivized in different locations or times.

---

## Author Response (AR1)

[revised manuscript text omitted]

**Commented [SZ9]:** RC#1 requested additional clarity regarding the method used to calculate non-beneficial and non-consumptive water fluxes within the main text. (Also, this revision is in response to RC#1 comment in section 3.1.)

[revised manuscript text omitted]